# Deficiency of Glucocerebrosidase Activity beyond Gaucher Disease: PSAP and LIMP-2 Dysfunctions

**DOI:** 10.3390/ijms25126615

**Published:** 2024-06-16

**Authors:** Eleonora Pavan, Paolo Peruzzo, Silvia Cattarossi, Natascha Bergamin, Andrea Bordugo, Annalisa Sechi, Maurizio Scarpa, Jessica Biasizzo, Fabiana Colucci, Andrea Dardis

**Affiliations:** 1Regional Coordinator Centre for Rare Diseases, University Hospital of Udine, 33100 Udine, Italy; eleonora.pavan@asufc.sanita.fvg.it (E.P.); paolo.peruzzo@asufc.sanita.fvg.it (P.P.); silvia.cattarossi@asufc.sanita.fvg.it (S.C.); natascha.bergamin@asufc.sanita.fvg.it (N.B.); andrea.bordugo@asufc.sanita.fvg.it (A.B.); annalisa.sechi@asufc.sanita.fvg.it (A.S.);; 2Clinical Pathology Division, Department of Laboratory Medicine, University Hospital Friuli Centrale ASUFC, 33100 Udine, Italy; jessica.biasizzo@asufc.sanita.fvg.it; 3Department of Neuroscience and Rehabilitation, University of Ferrara, 44121 Ferrara, Italy; clcfbn@unife.it

**Keywords:** glucocerebrosidase, LIMP-2, saposin C, prosaposin, Gaucher Disease, AMRF

## Abstract

Glucocerebrosidase (GCase) is a lysosomal enzyme that catalyzes the breakdown of glucosylceramide in the presence of its activator saposin C (SapC). SapC arises from the proteolytical cleavage of prosaposin (encoded by *PSAP* gene), which gives rise to four saposins. GCase is targeted to the lysosomes by LIMP-2, encoded by *SCARB2* gene. GCase deficiency causes Gaucher Disease (GD), which is mainly due to biallelic pathogenetic variants in the GCase-encoding gene, *GBA1*. However, impairment of GCase activity can be rarely caused by SapC or LIMP-2 deficiencies. We report a new case of LIMP-2 deficiency and a new case of SapC deficiency (missing all four saposins, PSAP deficiency), and measured common biomarkers of GD and GCase activity. Glucosylsphingosine and chitotriosidase activity in plasma were increased in GCase deficiencies caused by *PSAP* and *GBA1* mutations, whereas *SCARB2*-linked deficiency showed only Glucosylsphingosine elevation. GCase activity was reduced in fibroblasts and leukocytes: the decrease was sharper in *GBA1*- and *SCARB2*-mutant fibroblasts than *PSAP*-mutant ones; LIMP-2-deficient leukocytes displayed higher residual GCase activity than *GBA1*-mutant ones. Finally, we demonstrated that GCase mainly undergoes proteasomal degradation in LIMP-2-deficient fibroblasts and lysosomal degradation in PSAP-deficient fibroblasts. Thus, we analyzed the differential biochemical profile of GCase deficiencies due to the ultra-rare *PSAP* and *SCARB2* biallelic pathogenic variants in comparison with the profile observed in *GBA1*-linked GCase deficiency.

## 1. Introduction

Glucocerebrosidase (GCase), is a lysosomal enzyme belonging to the glycosyl hydrolase 30 family that catalyzes the breakdown of the glycosphingolipid Glucosylceramide (GlcCer) into ceramide and glucose.

The enzyme is encoded by the *GBA1* gene located on chromosome 1q21 (Gene Cards ID: GC01M159917), which generates two transcripts of 2.2 and 2.6 kb [1]. The *GBA1* mRNA presents two in-frame ATG start codons, which can both be used to produce a functional protein isoform—isoform long and isoform short [2,3]—of 536 and 516 amino acid residues, respectively. The longer isoform has a signal sequence encompassing residues 1 to 39 and has been chosen as the “canonical” sequence, while the shorter isoform, lacking the first 20 residues, presents a signal sequence encompassing only 19 residues. After the cleavage of the 39 and 19 leader sequences, both isoforms generate the same 497 residues protein with a predicted molecular weight of 55,598 Da [2].

The GCase enzyme is synthesized in the endoplasmic reticulum (ER) and undergoes co-translational glycosylations that are essential for catalytic activity in vivo, during transit through the Golgi [4,5,6]. In mammalian cells, lysosomal proteins are usually targeted to lysosomes through the recognition of their mannose-6-phosphate terminal residues by mannose-6-phosphate receptors (MPRs) [7]. However, GCase is trafficking to the lysosomal compartment in a mannose 6-phosphate-independent manner through its association with the lysosomal integral membrane protein type 2 (LIMP-2), a heavily N-glycosylated type III transmembrane protein encoded by *SCARB2* gene [8]. GCase and LIMP-2 interact in the pH-neutral environment of the ER to form a complex that passes through the Golgi and eventually reaches the lysosome, where acidic pH causes dissociation and subsequent release of active GCase [8].

In order to degrade lysosomal GlcCer, GCase requires the presence of negatively charged lipids and saposin C (SapC), the activator protein. SapC belongs to a family of four small lysosomal glycoproteins—SapsA, B, C, and D—all generated by the proteolytic processing of a common precursor, prosaposin, encoded by the *PSAP* gene [9,10,11].

GCase activity is essential for the normal catabolism of glycosphingolipids. Thus, its deficiency leads to the accumulation of GlcCer and other lipids within the lysosomes. The most frequent cause of GCase deficiency is the presence of biallelic pathogenic variants in the *GBA1* gene leading to Gaucher disease (GD), one of the most common lysosomal storage disorders [12,13,14]. The presence and severity/rate of progression of neurological involvement have been historically used as discriminating factors for GD classification into three different clinical phenotypes, although the clinical picture presents as a phenotypic continuum. Type 1 GD (MIM No. 230800) represents the most common phenotype, and it is characterized by enlargement of the liver and spleen, anemia, thrombocytopenia, and bone damage, leading to infarctions and fractures. Although type 1 GD is considered a non-neuronopathic form, there is increasing evidence of neurological manifestations (i.e., Parkinson’s syndrome, seizures, oligophrenia, perceptive deafness). Type 2 GD (MIM No. 230900) is a rare phenotype associated with an acute neurodegenerative course and death at a very early age. Finally, type 3 GD, the chronic neuronopathic GD (MIM No. 231000), comprises an extremely heterogeneous group of patients with either mild or severe systemic disease associated with some form of neurological involvement; the onset of symptoms might range from childhood to early adulthood.

Very rarely, deficiency of GCase activity can be caused by biallelic pathogenic variants in the *PSAP* gene, leading to a deficiency in the GCase activator SapC or, in the *SCARB2* gene, causing the deficiency of the GCase transporter LIMP-2. While patients affected by SapC deficiency display typical GD clinical phenotype, patients with biallelic pathogenic variants in the *SCARB2* gene present action myoclonus renal failure syndrome (AMRF, MIM No. 254900), a condition that shares some neurological clinical features with type 3 GD, such as myoclonic epilepsy. However, AMRF-affected patients do not present the characteristic hematological and visceral manifestations of GD3, strongly suggesting that in some tissues—in particular in blood cells—GCase may be transported to the lysosomes in a LIMP-2-independent mechanism [15].

In this paper, we present an in-depth characterization of biochemical and cellular features of patients presenting a deficiency of GCase activity due to the ultra-rare presence of biallelic pathogenic variants in *PSAP* or *SCARB2* genes, comparing them to *GBA1*-linked GCase deficiency, highlighting the characteristic laboratory features that differentiate these disorders.

## 2. Results

In order to compare the biochemical and cellular features of patients presenting with a deficiency of GCase activity due to biallelic pathogenic variants in *GBA1*, *PSAP*, or *SCARB2* genes, we analyzed two AMRF patients (LIMP2_PT1 already described by Dardis and colleagues [15,16], and LIMP2_PT2 presented below as Case 1), one patient with PSAP deficiency presented below as Case 2 (PSAP_PT), and 8 GD patients presenting *GBA1* biallelic pathogenic variants (GBA1_PTs—Appendix A).

### 2.1. Case Reports

Case report 1 (LIMP2_PT2) presented during pregnancy when she was 34 years old, with upper extremity limbs action myoclonic jerks, triggered by movements and exacerbated by anxiety and auditory stimuli, associated with mild proteinuria.

In the following years, proteinuria increased, and renal failure occurred; action myoclonus worsened, also affecting sitting, walking, eating, and speech. Indeed, she used walking aids, and her husband helped her cut food and eat.

At the age of 37 years, molecular analysis by NGS proved homozygous for the known pathogenic variant c.1087C>A, p.(H363N) in the *SCARB2* gene, confirming the diagnosis of action myoclonus renal failure (AMRF).

Case report 2 (PSAP_PT) is a 4-month-old baby presented with hepatosplenomegaly and severe neurological disease characterized by hypotonia, nystagmus, and swallowing difficulty. The diagnosis of GD type 2 was hypothesized. Chitotriosidase activity was measured as a screening test and resulted significantly elevated (in the range of GD-affected patients). The *GBA1* gene was then analyzed by specific PCR amplification followed by Sanger sequencing and multiplex ligation-probe amplification (MLPA); however, no genetic variants were identified. Considering this result, a PSAP deficiency was suspected, and the diagnosis was confirmed detecting the already described pathogenic variant, c.889G>T, p.(E297*) in the *PSAP* gene, in homozygosis. The analysis of the mRNA expressed in fibroblasts in the presence or absence of anysomicin, which inhibits nonsense-mediated decay, showed that the mutant transcript is degraded by this process (Appendix A). The genotype was confirmed in the child’s parents, who were healthy carriers of the identified pathogenic variant. A few months after diagnosis the patient died.

### 2.2. Plasma GD Biomarkers

We measured the plasma levels of the most widely used biomarkers of GD: chitotriosidase activity, a marker of macrophage activation, and glucosylsphingosine (GlcSph), the deacylated forms of the Glucosylceramide. As shown in Table 1, chitotriosidase was highly increased in PSAP_PT and GBA1_PTs, while it was normal or slightly increased in LIMP2_PTs. All patients presented increased levels of GlcSph in plasma.

### 2.3. Other Plasma Biomarkers

As we routinely perform multiplex assessment of glycosphingolipid biomarkers, we analyzed the plasma levels of Globotriaosylsfingosine (Lyso-Gb3) and N-palmitoyl-O-phosphocholineserine (PPCS) in both LIMP-2 and PSAP deficient patients. As shown in Table 2, PSAP_PT presented increased levels of Lyso-Gb3 and PPCS.

### 2.4. GCase Activity

Second, we compared the GCase activity in plasma, fibroblasts, and leukocytes whenever available. While, as expected, the activity in plasma was undetectable in patients carrying biallelic variants in *GBA1* (GBA1_PTs; n = 5, Appendix A) and the PSAP_PT, it was increased in those carrying biallelic variants in *SCARB2* (LIMP2_PTs) (Table 3). All patients presented deficient GCase activity in cells. However, the residual activity in leukocytes was higher in LIMP2_PTs than in GBA1_PTs (n = 5, Appendix A), whereas fibroblasts of LIMP2_PTs and GBA1_PTs (n = 4, Appendix A) presented comparable results. Unfortunately, the GCase activity in the leukocytes from the PSAP_PT was not available; however, the residual activity in fibroblasts was higher than the activity present in GBA1_PTs and LIMP2_PTs (Table 3).

Regarding LIMP2_PTs, the reduction of GCase activity in cells and the presence of activity in plasma were expected, as previous studies in cells from LIMP2_PT1 had already shown that reduced GCase intracellular activity was due to the partial degradation of the enzyme in the Endoplasmic reticulum via proteasome and the concomitant increased secretion of the enzyme [15,16]. On the contrary, the reduction of GCase activity in fibroblasts from PSAP_PT was quite unexpected since SapC is not needed for the in vitro action of GCase using the synthetic 4MU substrate.

### 2.5. GCase Protein Expression in Fibroblasts

To confirm data obtained in the LIMP2_PT1 and to better understand the causes leading to the detection of low in vitro levels of GCase activity in cells from the PSAP_PT, we analyzed the levels of GCase protein expression and ER-to-Golgi transition of GCase in patient’s fibroblasts by digesting cells with endoglycosidase H (Endo H) or endoglycosidase F (Endo F). Endo H specifically cleaves high mannose (>4 mannose residues) but not mature N-glycan complexes, allowing differentiation between immature glycoproteins that have not reached the mid-Golgi (Endo H-sensitive) and mature glycoproteins (Endo H-resistant). By removing all aspargine-linked glycans, Endo F serves as a positive control for glycoprotein digestion (despite their mature status) and migration to SDS-PAGE.

As shown in Figure 1A, in both LIMP2_PT1 and LIMP2_PT2 the abundance of GCase was decreased compared with wild-type (wt) cells, and the protein was completely retained in the ER. Furthermore, treatment of cells with the proteasomal inhibitor MG132 resulted in a partial rescue of GCase protein abundance (Figure 1B,C).

In PSAP_PT cells, the levels of GCase protein were also decreased compared with wt cells. However, although a small portion of GCase protein was retained in the ER, it is clear that a fair amount of protein reaches the lysosome (Figure 1D). To test whether the GCase protein that reaches the lysosome was further subjected to lysosomal degradation, we analyzed the GCase abundance in the presence of lysosomal protease inhibitor cocktail (lPIC). As shown in Figure 1E,F, the inhibition of lysosomal degradation resulted in a significant rescue of GCase protein abundance. Taken together, these results confirm data already reported showing that in LIMP-2-deficient cells, GCase is completely retained in the ER and subjected to proteasomal degradation [15,17] and demonstrate that in PSAP-deficient cells, low GCase protein levels are at least in part due to increased degradation within the lysosome.

### 2.6. Cholesterol Accumulation in Fibroblasts

The elevation in plasma PPCS levels, used as a biomarker of Niemann–Pick C disease (NPCD), in PSAP_PT prompted us to hypothesize an impairment in cholesterol homeostasis in this condition. Although normal levels of PPCS were identified in LIMP2_PT2, a role of LIMP-2 in cholesterol trafficking alongside NPC1 and NPC2 has recently been proposed. Based on this evidence, we assessed the intracellular cholesterol storage in LIMP-2- and PSAP-deficient fibroblasts by filipin staining. Fibroblasts from an NPC patient were used as positive control. As shown in Figure 2 and Appendix A, even though cholesterol accumulation was not as pronounced as in NPCD fibroblasts, both LIMP2_PTs and PSAP_PT fibroblasts accumulate a fair amount of this lipid. Interestingly, PSAP_PT showed a more pronounced accumulation than LIMP2_PTs; in addition, cholesterol storage seems to be higher in LIMP2_PT2 in comparison with LIMP2_PT1. These observations suggest an impairment of cholesterol metabolism in PSAP and LIMP-2 deficiencies.

## 3. Discussion

In this paper, we highlighted the characteristic laboratory features that differentiate the ultra-rare deficiencies of GCase activity due to biallelic pathogenic variants in PSAP or LIMP-2 encoding genes from the more common GCase deficiency due to biallelic pathogenic variants in *GBA1* gene.

To the best of our knowledge, 35 families affected by LIMP-2 deficiency and 9 affected by PSAP deficiency have been described so far. Among them, only 8 LIMP-2-deficient and 7 PSAP-deficient patients have been assessed for GCase activity and/or plasma GD biomarkers (Table 4 and Table 5), including the present study.

In all cases of patients affected by LIMP-2 deficiency in which GCase activity was measured, it resulted to be strongly reduced in fibroblasts [16,17,21,53], increased in plasma [16,53], and slightly reduced or normal in leukocytes/lymphocytes [16,17,21,22,23,53]. No changes in chitotriosidase activity were reported [16,17,23], and increased levels of GlcSph were identified in plasma and fibroblasts [23,53]. Finally, the whole amount of GCase expressed by LIMP-2 deficient fibroblasts was immature and retained in the ER [15,17].

The biochemical features of LIMP2_PT2 fit in this picture being consistent with previously reported observations. It is worth noting that although the plasma level of GlcSph in this patient is more than 10 times higher than the levels found in healthy controls, it was below the range found in GD patients due to *GBA1* biallelic pathogenic variants. Assuming that the GlcSph detected in plasma is mainly released by the cells from the monocyte-macrophage system, this finding would be consistent with the presence of a quite high residual activity of GCase detected in blood cells, which would prevent massive accumulation of substrate.

Taken together, these data and those obtained using different cellular and animal models of LIMP-2 deficiency [15,53,54] strongly suggest that while in fibroblasts and neurons, GCase targeting to the lysosomes is completely dependent on LIMP-2, in blood cells, GCase is partially targeted to lysosomes by a LIMP-2-independent mechanism; even though the existence of a putative secondary mechanism has been suggested, further studies are needed to identify the secondary transporter.

Compared with previously described patients, LIMP2_PT2 presents a higher reduction of GCase activity in leukocytes and a slight increase of plasma chitotriosidase activity, indicating some degree of macrophage activation. Two different hypotheses might explain these findings. On one hand, it is possible that in this patient the LIMP-2 independent mechanism for lysosomal sorting of GCase in blood cells would be less active. On the other hand, these features could be associated with the nature of the *SCARB2* pathogenic variant identified in the patient. Indeed, she was homozygous for the missense variant p.(H363N). This is the only missense variant described in patients affected by AMRF, while all other pathogenic variants, including the one present in one allele of LIMP2_PT1, are splicing, nonsense, or small deletions/insertions leading to the generation of premature stop codons (Table 4), likely causing the degradation of the expressed mRNA, and/or the expression of truncated proteins, most probably unable to bind GCase [15,21,26,40,43,55]. Conversely, it has been demonstrated that the p.H363N LIMP-2 mutant protein is retained in the ER and binds GCase with a higher affinity compared with the wt protein [55]. Considering this scenario, it is possible to hypothesize that in blood cells carrying this particular variant in homozygosis, a higher amount of GCase remains associated with LIMP-2 in the ER, leaving less GCase available to be delivered to the lysosome by the LIMP-2 independent pathway.

Recently, LIMP-2 was reported to act in parallel with NPC1-NPC2, mediating a secondary lysosomal cholesterol efflux [56]. Thus, it is not surprising that LIMP-2 deficient fibroblasts do accumulate a fair amount of this lipid. The magnitude of this increase is lower than the one observed in PSAP deficiency: this may explain why no changes in PPCS levels in plasma of LIMP2_PTs were recorded. As both LIMP2_PT1 and LIMP2_PT2 carry the same pathogenic variant in the *SCARB2* gene, the reason beyond the slightly different amount of cholesterol storage observed in the two patients cannot be due to the features of the mutant protein itself: this may suggest a certain grade of interpersonal variability.

As mentioned above, the GCase activator SapC derives from the prosaposin precursor PSAP, which is encoded by the *PSAP* gene, synthesized in the ER, transported to the Golgi, and eventually trafficked to the lysosome, where it is cleaved into the four saposins: SapA, SapB, SapC, and SapD [9,11]. These four proteins act as activators for several enzymes (e.g., SapA works as activator for galactoserebrosidase, SapB is the activator of arylsulfatase A, SapC is the activator of GCase, and SapD is the activator of the acid ceramidase). Thus, an impaired GCase activity in vivo due to a deficiency of its activator could be caused by *PSAP* biallelic pathogenic variants affecting the expression of the PSAP precursor and therefore resulting in the absence of all saposins ([44,45,46,47,48,49,50,51,52], and present study), or by missense variants in the SapC domain, affecting the sole SapC function [57,58,59,60,61,62,63,64,65,66,67,68].

The patient described here presented a nonsense variant in homozygosis, leading to a degradation of the *PSAP* mRNA by nonsense-mediated decay and thus resulting in a deficiency of all four Saps.

A revision of the literature showed that the in-vitro levels of GCase activity were found to be significantly reduced in cells from all PSAP-deficient patients in which this enzyme has been measured [44,45,46,49,51,52]. In general, the detected residual activity was higher than the activity usually found in patients affected by GD due to *GBA1* biallelic pathogenic variants. In addition, accumulation of GlcCer or its deacylated form GlcSph and Globotriaosylceramide (Gb3) or its deacylated form Lyso-Gb3 were observed [44,45,46,47,49,52]. Thus, the PSAP-deficient patient described here presented the same biochemical profile as previously reported cases.

Furthermore, we found increased levels of PPCS and increased activity of chitotriosidase (reflecting the macrophage activation), and non-detectable levels of GCase in plasma. All these parameters might be of use for differential diagnosis.

Once again, it is worth noting that although the plasma levels of GlcSph in the PSAP_PT are higher than the levels found in healthy controls, they were below the range found in GD patients due to *GBA1* pathogenic variants. However, this result is not unexpected considering that GlcSph is generated from GlcCer by the action of the acid ceramidase which requires the activator SapD, a protein lacking in patients with complete PSAP deficiency.

Patients missing only SapC share some biochemical features with PSAP-deficient patients. For instance, accumulation of GlcCer or its deacylated form GlcSph was observed whenever assessed [57,58,59,60,61,62,65,68]. Moreover, these patients present an increased activity of chitotriosidase [63,65,67]. However, these patients are not expected to accumulate other glycosphingolipids and, probably being SapD-unaffected, would accumulate higher levels of GlcSph compared to PSAP-deficient patients. Further studies are needed to confirm this hypothesis.

The reduction of GCase activity in cells from PSAP-deficient patients is quite intriguing since it is well known that SapC is not needed for the degradation of the artificial substrate used to measure the GCase activity in vitro. However, our data suggest that the in vitro reduction of GCase activity found in PSAP-deficient patients is likely to be due to reduced levels of GCase protein [52]. Furthermore, our data suggest that this reduction is mainly caused by increased levels of lysosomal degradation of GCase. Since SapC and GCase interact within the lysosome, it seems reasonable to hypothesize that SapC is needed to prevent GCase lysosomal degradation [69].

In line with the increased plasma levels of PPCS, we observed lysosomal accumulation of unesterified cholesterol. Lysosomal accumulation of cholesterol was already reported in four SapC-deficient patients [64]: thus, it is not surprising to observe the storage of this lipid in PSAP-deficient cells as well. The reason beyond this observation may rely on the role of Saps as cholesterol transporters: these small molecules were indeed reported to form complexes with many lipids, including cholesterol; in particular, SapA seemed to act similarly to HDL [70,71,72,73] and was reported to deliver cholesterol to LIMP-2 [56].

Taking together our findings and those already reported in the literature, it is possible to depict a model in which the mechanism leading to these characteristic laboratory features that differentiated deficient GCase activity disorders is highlighted (Figure 3).

In conclusion, the data presented here could be useful for the differential diagnosis of these conditions and the development of specific therapeutic strategies.

## 4. Materials and Methods

### 4.1. Patients

Plasma, DNA, leukocytes, and/or fibroblasts from two patients affected by AMRF due to biallelic variants in the *SCARB2* gene (LIMP2_PT1; LIMP2_PT2), one patient affected by prosaposin deficiency (PSAP_PT) and 8 patients affected by the neuronopathic phenotype of GD due to biallelic *GBA1* variants (3 GD3 and 5 GD2 = GBA1_PTs) were obtained during the diagnostic workup at the Regional Coordinator Centre for Rare Diseases, Udine. Clinical and genetic characteristics of LIMP2_PT1 were previously reported [16], while those from LIMP2_PT2 and PSAP_PT are described above. Informed consent was obtained from all subjects.

### 4.2. Cell Culture, MG132, lPIC, and Anisomycin Treatments

Primary human fibroblasts were cultured and maintained in Dulbecco’s modified Eagle’s medium High glucose (EuroClone, Pero, Italy) containing 10% fetal bovine serum (Gibco-Thermo Fisher, Waltham, MA, USA), and 1% penicillin/streptomycin (Sigma, St. Louis, MO, USA) in a humidified atmosphere containing 5% CO_2_ at 37 °C.

To assess GCase proteasomal degradation, 300 × 10^3^ fibroblasts were grown in a T25 flask and treated for 96 h with 0.2 µM MG132 (Sigma, St. Louis, MO, USA) dissolved in dimethyl sulfoxide (DMSO—Sigma, St. Louis, MO, USA) or with an equal volume of vehicle (DMSO) as a negative control (treatment was repeated every other day).

To assess GCase lysosomal degradation, 300 × 10^3^ fibroblasts were grown in a T25 flask and treated for 48 h with a lysosomal proteases inhibitor cocktail consisting of 50 µM chymostatin (Sigma, St. Louis, MO, USA) dissolved in DMSO, 50 µM Leupeptin (Sigma, St. Louis, MO, USA) dissolved in water, 10 µM Pepstatin (Sigma, St. Louis, MO, USA) dissolved in DMSO, E64d 5 µM (Sigma, St. Louis, MO, USA) dissolved in DMSO, or with an equal volume of vehicle (DMSO), as negative control (treatment was repeated every 24 h).

To inhibit nonsense-mediated decay, primary fibroblasts were treated with 100 µg/mL Anisomycin (Sigma, St. Louis, MO, USA) for 5 h.

### 4.3. Filipin Staining

To assess primary fibroblasts for cholesterol accumulation, cells were plated on glass coverslips and starved for 48 h. LDL (L3486, Invitrogen, Waltham, MA, USA) were subsequently provided to the cells, and after 24 h, the staining was performed as previously described [78]. Briefly, cells were washed with PBS, fixed with 3% paraformaldehyde (Sigma, St. Louis, MO, USA) for 30 min, rinsed with PBS, incubated with 1.5 mg/mL (Sigma, St. Louis, MO, USA) for 10 min, and eventually stained with filipin (0.05 mg/mL in PBS 10% FBS, Sigma, St. Louis, MO, USA) for 2 h. Cells were visualized, and images were obtained with a live cell imaging dedicated system consisting of a Leica DMI 6000B microscope connected to a Leica DFC350FX camera (Leica Microsystems, Wetzlar, Germany).

### 4.4. Protein Extraction, Western Blot, Endo H, and Endo F Digestion

In order to evaluate expression levels of GCase, cells were pelleted, lysed in cell lysis buffer TNN (Tris-HCl 100 mM pH 8, NaCl 250 mM, NP40 0.5%), sonicated, and centrifugated (10 min at 4 °C at 14,000 rpm). Supernatants representing the protein extracts were quantified for protein content using the Biorad-Protein Assay (BioRad, Hercules, CA, USA).

Upon denaturing the samples for 5 min at 95 °C, 30 µg of protein extracts were separated on a 4–20% gradient Mini-Protean TGX pre-cast gel (BioRad, Hercules, CA, USA) in running buffer (Running Buffer 10 X: Tris 25 mM, Glycine 0.191 M, SDS 0.1% *w*/*v*). Fractionated proteins were transferred to nitrocellulose membrane (BioRad, Hercules, CA, USA) in transfer buffer (Tris 25 mM, Glycine 0.189 M, 20% MeOH), and membranes were blocked in 5% Blotting-Grade Blocker (BioRad, Hercules, CA, USA) in PBS-T (0.1% Tween 20 in PBS) for 2 h. Then, membranes were incubated overnight at 4 °C with the appropriate primary antibody 1:1000 (GBA 2E2 (H00002629-M01—Abnova, Taipei City, Taiwan), actin (A2066, Sigma, St. Louis, MO, USA), and then washed, incubated with the appropriate secondary antibody (anti-mouse HRP A9044—Sigma, St. Louis, MO, USA; anti-rabbit 31460—Invitrogen, Waltham, MA, USA) for 1 h at RT, and developed with SuperSignal West Pico reagents (Thermo Fisher Scientific, Waltham, MA, USA). Blots were quantified by using a Uvitech Cambridge (UVITEC, Cambridge, UK).

To identify the mature GCase fraction, 30 µg of total protein deriving from treated and untreated fibroblasts was digested at 37 °C overnight with 2.5 µL of Endo H or 1 µL of Endo F (Roche, Basel, Switzerland) in the appropriate buffer (Endo H Buffer: Buffer sodium citrate 50 mM pH 5.2, PMSF 0.5 mM, SDS 0.1%, Triton X 100 0.5%, Mercaptoethanol 0.1 M; Endo F Buffer: Buffer sodium phosphate 100 mM pH 7.2, EDTA 10 mM, Triton X 100 0.1%, SDS 0.1%, Mercaptoethanol 1%) and then processed as described above.

### 4.5. DNA Extraction, Amplification, and Sequencing

DNA was extracted using DNeasy blood and tissue kit (Qiagen GmbH, Hilden, Germany) according to the manufacturer’s protocols. *SCARB2* and *GBA1* genes were amplified and sequenced as previously described [15,16]. Exon and exon-flanking regions of *PSAP* gene were amplified using the primers reported in Appendix A according to the following protocol: 95 °C 5 min; 35 cycles consisting of 95 °C 30 s, 62 °C 30 s, 72 °C 30 s; 72 °C 7 min. PCR products were purified using Exo-Prostar (Cytiva, Marlborough, MA, USA); BigDye (Applied Biosystem-Thermo Fisher Scientific, Waltham, MA, USA), and each primer reported in Appendix A was used in sequencing reaction (26 cycles consisting of 95 °C 10 s, 50 °C 15 s, 62 °C 2 min). Sequences were purified and loaded in a 3500xL Genetic Analyzer (Applied Biosystems-Thermo Fisher Scientific, Waltham, MA, USA). Sequencing analysis was performed using Chromas software (version 2.6.6, Technelysium Pty Ltd., South Brisbane, QLD, Australia). Accession number of *PSAP* RNA sequence: NM_002778.

### 4.6. RNA Extraction and PSAP cDNA Amplification

Total RNA was isolated using the QIAShredder and the RNeasy mini kit (Qiagen, Hilden, Germany). First-strand cDNA synthesis was performed with 2 μg total RNA using Superscript III Reverse Transcriptase (Invitrogen-Thermo Fisher Scientific, Waltham, MA, USA) according to manufacturer’s instructions. *PSAP* cDNA was amplified using PSAP RNA F and PSAP RNA R primers (Appendix A) according to the following protocol: 95 °C 5 min; 35 cycles consisting of 95 °C 30 s, 60 °C 30 s, 72 °C 1 min; 72 °C 7 min.

### 4.7. GCase Enzymatic Activity

GCase activity was measured in plasma, leukocytes, and fibroblasts using the fluorigenic substrate 4-methylumbelliferyl-β-D-glucopyranoside (M3663, Sigma-Aldrich, St. Louis, MO, USA).

Leukocytes were isolated from the peripheral blood, and proteins were extracted and quantitated by Lowry Assay.

As regards GCase activity in plasma and leukocytes, 10 µL of plasma or 50 µg of proteins in 40 µL was incubated with 10 μL of TDC 2% and 100 µL of substrate 10 mM in Citrate (0.1 M)/Phosphate (0.2 M) pH 5.2 buffer at 37 °C for 2 h. Carbonate buffer 0.5 M pH 10.7 was used to stop the reaction the fluorescent product was quantified using a fluorimeter (SPECTRAmax Gemini XPS, Molecular Devices, San Jose, CA, USA) at an excitation wavelength of 365 nm and emission of 495 nm.

As regards GCase activity in fibroblasts, 10 µL containing 10 µg of proteins was incubated with 10 μL of substrate 5 mM in acetate buffer 0.1 M pH 4.2 at 37 °C for 3 h. Carbonate buffer 0.5 M pH 10.7 was used to stop the reaction the fluorescent product was quantified using a fluorimeter (SPECTRAmax Gemini XPS, Molecular Devices, San Jose, CA, USA) at an excitation wavelength of 365 nm and emission of 495 nm.

### 4.8. Chitotriosidase Activity

Plasma chitotriosidase activity was determined using the fluorigenic substrate 4-Methylumbelliferyl β-D-N,N′,N″-triacetylchitotrioside (M5639, Sigma-Aldrich, St. Louis, MO, USA); 5 µL plasma of was incubated with 100 μL of substrate 0.022 mM in Citrate (0.1 M)/Phosphate (0.2 M) pH 5.2 buffer at 37 °C for 15 min. Carbonate buffer 0.5 M pH 10.7 was used to stop the reaction the fluorescent product was quantified using a fluorimeter (SPECTRAmax Gemini XPS, Molecular Devices, San Jose, CA, USA) at an excitation wavelength of 365 nm and emission of 495 nm.

### 4.9. Glucosylsphingosine (GlcSph), Globotriaosylsphingosine (Lyso-Gb3) and N-Palmitoyl-O-phosphocholineserine (PPCS) Accumulation

GlcSph, Lyso-Gb3, and PPCS in plasma were measured by LC-MS/MS as previously described [79]. Briefly, after protein precipitation, evaporation, and reconstitution in mobile phase, reverse-phase liquid chromatography was performed using a Shimadzu Nexera CL UHPLC (Shimadzu, Kyoto, Japan) and a Poroshell 120 EC-C8 column, 3.0 × 50.0 mm with 2.7 μm particle size (Agilent, Santa Clara, CA, USA). Mass spectrometry detection was carried out with AB Sciex 6500 QTrap tandem mass spectrometer (Sciex, Framingham, MA, USA) set in positive mode using an electrospray ionization (ESI). D5-glucosylsphingosine, D7-Lyso-Gb3, and D9-Lyso-SM, were used as internal standards for GlcSph, Lyso-Gb3, and PPCS, respectively.

### 4.10. Statistical Analysis

Statistical significance was determined by Student’s *t*-test; *p*-value < 0.05 was considered statistically significant.

## Figures and Tables

**Figure 1 ijms-25-06615-f001:**
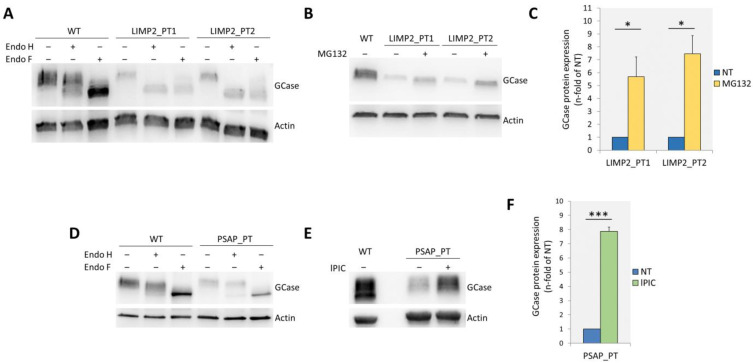
(**A**) Western blot analysis of GCase expression in WT, LIMP2_PT1, and LIMP2_PT2 fibroblasts treated (+) or not (−) with Endo H or Endo F; (**B**) western blot analysis of GCase expression in WT, LIMP2_PT1, and LIMP2_PT2 fibroblasts treated (+) or not (−) with the proteasomal inhibitor MG132; (**C**) quantitation of GCase expression in LIMP2_PT1, and LIMP2_PT2 fibroblasts treated with the proteasomal inhibitor MG132; (**D**) western blot analysis of GCase expression in WT and PSAP_PT fibroblasts treated (+) or not (−) with Endo H; (**E**) western blot analysis of GCase expression in WT and PSAP_PT fibroblasts treated (+) or not (−) with the lysosomal proteases inhibitor cocktail lPIC; (**F**) quantitation of GCase expression of PSAP_PT fibroblasts treated with lPIC. Results are expressed as mean ± SD of three independent experiments. * *p*-value < 0.05; *** *p*-value < 0.001.

**Figure 2 ijms-25-06615-f002:**
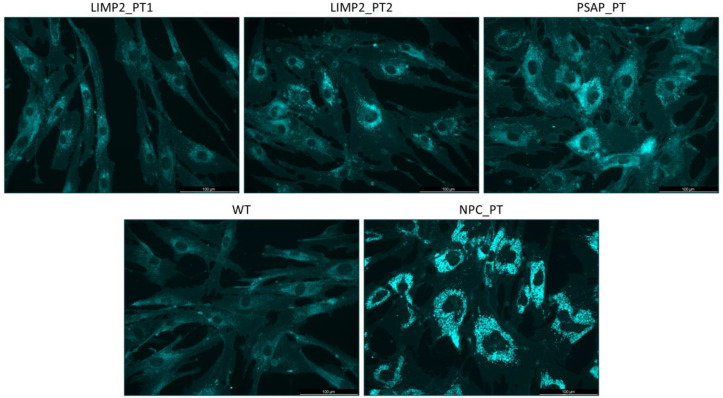
Filipin stained intracellular unesterified cholesterol of LIMP2_PT1, LIMP2_PT2, PSAP_PT fibroblasts, fibroblasts from a healthy control (WT), and Niemann–Pick type C-affected patient (NPC). Scale bar 100 µm.

**Figure 3 ijms-25-06615-f003:**
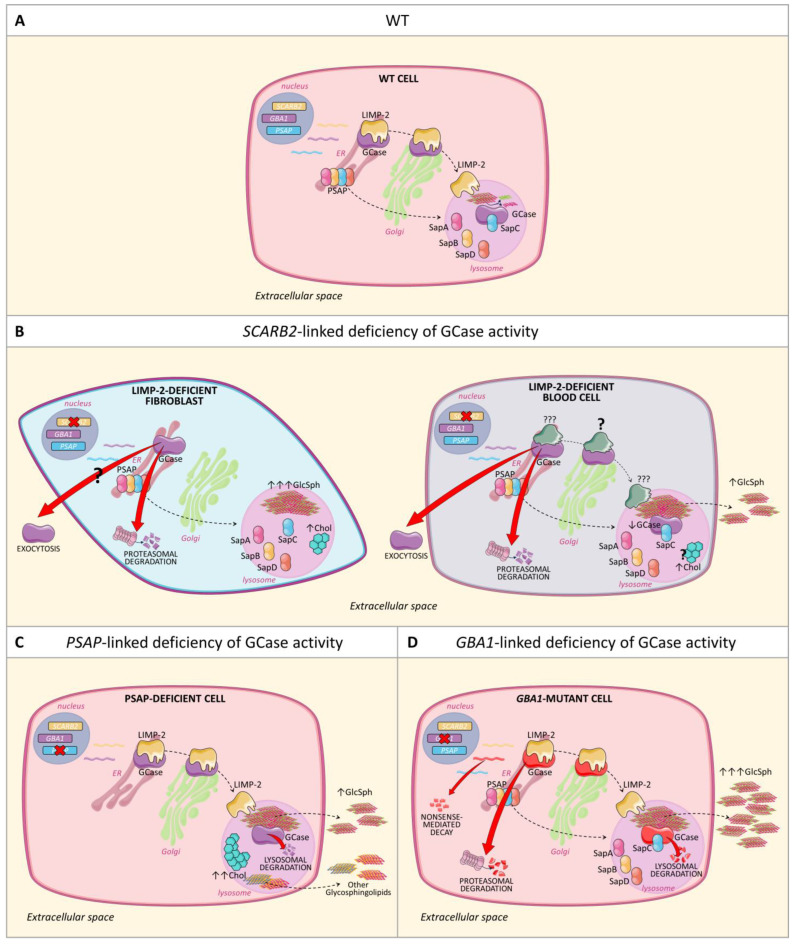
GCase fate in a normal cell and in cells presenting GCase-deficient activity due to biallelic pathogenic variants of *SCARB2*, *PSAP*, or *GBA1* genes. (**A**) In a normal cell, GCase is synthesized in the ER, subsequently processed in the Golgi, and eventually delivered to the lysosome by its transporter LIMP-2. In the lysosome, GCase catalyzes the degradation of GlcCer in the presence of its activator SapC. (**B**) In *SCARB2*-linked deficiency of GCase activity, GCase fate differs according to the cell type. In LIMP-2-deficient fibroblasts, wt GCase cannot reach the lysosome and is degraded via proteasome and possibly also released outside the cell ([15,17,53] and present study); consequently, GlcSph accumulates within lysosomes ([53]). In LIMP-2-deficient blood cells, wt GCase undergoes proteasomal degradation, is released outside the cells and, in small amounts, also somehow reaches the lysosome, where it seems able to degrade GlcCer to some extent, resulting in moderate plasma release of GlcSph ([15,16,17,21,22,23,53], and present study). In addition, due to the lack of one of its transporter LIMP-2, lysosomal cholesterol (Chol) efflux is partially impaired ([64], and present study). (**C**) In *PSAP*-linked deficiency of GCase activity, wt GCase can effectively reach the lysosome but is unable to actively degrade GlcCer, as SapC is missing. Thus, the wt GCase is degraded by lysosomal proteases and GlcCer and GlcSph are accumulating ([47,49,52], and present study). Likely due to the lack of SapD, low plasma levels of GlcSph are observed in comparison with *GBA1*-linked deficiency of GCase activity. In addition, since all Saps are lacking, other glycosphingolipids and cholesterol (Chol) are accumulating ([64] and present study). (**D**) In GD cells, *GBA1* biallelic pathogenic variants lead to the progressive accumulation of GlcCer and its deacylated form GlcSph which is abundantly released outside the cells resulting in high plasma levels of this glycosphingolipid. According to the type of *GBA1* variant, the loss of GCase activity may depend on nonsense-mediated decay of mutant *GBA1* mRNA, proteasomal degradation of ER-retained mutant GCase protein, lysosomal degradation of mutant GCases that are delivered to lysosomes ([74,75,76,77]). Parts of the figures were drawn using pictures from Servier Medical Art. ServierMedical Art by Servier is licensed under a Creative Commons Attribution 3.0 Unported License (https://creativecommons.org/licenses/by/3.0/, accessed on 14 April 2024).

**Table 1 ijms-25-06615-t001:** Plasma GD biomarkers. Abbreviations: NA = non-available.

	Chitotriosidase Activity (Range: nmol/mL/h)	GlcSph (Range: ng/mL)
Healthy controls (n = 100)	21.2–137.1	0.1–1.8
GBA1_PTs	977.0–12,137.0	27.3–126.7
LIMP2_PT1	25.0	NA
LIMP2_PT2	186.8	23.3
PSAP_PT	2951.0	19.0

**Table 2 ijms-25-06615-t002:** Other plasma biomarkers. Abbreviations: NA = non-available; ND = non-detectable.

	Lyso-Gb3 (Range: ng/mL)	PPCS (Range: ng/mL)
Healthy controls (n = 100)	ND–0.87	9.8–200.9
GBA1_PTs	0.17–0.74	19.0–100.1
LIMP2_PT1	NA	NA
LIMP2_PT2	0.40	35.2
PSAP_PT	1.98	745.9

**Table 3 ijms-25-06615-t003:** GCase activity. Abbreviations: NA = non-available; ND = non-detected.

	Leukocytes		Fibroblasts		Plasma
	(nmol/mg/h) Mean ± SD [Range]	% of Healthy Controls	(nmol/mg/h) Mean ± SD [Range]	% of Healthy Controls	(nmol/mL/h)
Healthy controls *	17.6 ± 2.3 [15.4–20.8]	100	127.4 ± 34.1 [97.9–179.9]	100	ND
GBA1_PTs	1.0 ± 0.3 [0.6–1.4]	5.7	4.9 ± 3.1 [1.7–7.6]	3.8	ND
LIMP2_PT1	3.9	22.2	1.7	1.3	10.5
LIMP2_PT2	2.2	12.5	5.2	4.1	27.4
PSAP_PT	NA	NA	19.7	15.5	ND

* Leukocytes: n = 20; fibroblasts: n = 15; plasma n = 20.

**Table 4 ijms-25-06615-t004:** LIMP-2 deficiency in literature. Reference transcript: NM_005506.4; reference protein: NP_005497.1. Description of DNA and protein sequence variants are indicated according to HGVS nomenclature guidelines. Abbreviations: NA = non-assessed, F = fibroblasts, LC = leukocytes, LP = lymphocytes; P = plasma; chito = chitotriosidase activity.

Family	N° of Patients	Pathogenic Variant	Predicted Protein	Type of Variant	GCase Activity	Plasma GD Biomarkers	References
1	2	c.533G>A/c.533G>A	p.(W178*)/p.(W178*)	Nonsense	Reduced (F), normal (LC), increased (P)	Increased (GlcSph); normal (chito)	[17,18]
2	1	c.1087C>A/c.424-2A>C	p.(H363N)/p.?	missense/splicing	Reduced (F), slightly reduced (LC), increased (P)	Increased (GlcSph); normal (chito)	[15,16,19,20], present study
3	1	c.1270C>T/c.1270C>T	p.(R424*)/p.(R424*)	Nonsense	Reduced (F), slightly reduced (LP), increased (P)	NA	[21]
4	1	c.434_435dup/c.862C>T	p.(W146Sfs*161)/p.(Q288*)	frameshift/nonsense	Reduced (LC)	NA	[22]
5	1	c.704+1G>A/c.704+1G>A	p.?/p.?	Splicing	Reduced (LC)	Slightly increased (GlcSph); normal (chito)	[23]
6	3	c.862C>T/c.862C>T	p.(Q288*)/p.(Q288*)	Nonsense	NA	NA	[24,25]
7	1	c.434_435dup/c.434_435dup ^§^	p.(W146Sfs*161)/p.(W146Sfs*161)	Frameshift	NA	NA	[24,25]
8	1	c.1239+1G>T/c.1239+1G>T	p.?/p.?	Splicing	NA	NA	[25]
9	1	c.296del/c.704+5G>A	p.(N99Ifs*34)/p.?	frameshift/splicing	NA	NA	[25,26]
10	1	c.434_435dup/c.434_435dup	p.(W146Sfs*161)/p.(W146Sfs*161)	frameshift	NA	NA	[25]
11	3	c.111del/c.111del	p.(I37Mfs*7)/p.(I37Mfs*7)	frameshift	NA	NA	[24,27]
12	4	c.704+1G>A/c.704+1G>A	p.?/p.?	splicing	NA	NA	[28,29]
13	1	c.434_435dup/c.434_435dup	p.(W146Sfs*161)/p.(W146Sfs*161)	frameshift	NA	NA	[26]
14	1	c.434_435dup/c.434_435dup	p.(W146Sfs*161)/p.(W146Sfs*161)	frameshift	NA	NA	[26]
15	5	c.134del/c.134del	p.(N45Mfs*88)/p.(N45Mfs*88)	frameshift	NA	NA	[30]
16	2	c.134del/c.134del	p.(N45Mfs*88)/p.(N45Mfs*88)	frameshift	NA	NA	[31]
17	1	c.104del/c.104del	p.(Q35Rfs*9)/p.(Q35Rfs*9)	frameshift	NA	NA	[32]
18	1	c.956del/c.956del	p.(L319Rfs*6)/p.(L319Rfs*6)	frameshift	NA	NA	[33]
19	2	c.423+1G>A/c.423+1G>A	p.?/p.?	splicing	NA	NA	[34]
20	2	c.704+1G>A/c.704+1G>A	p.?/p.?	splicing	NA	NA	[35]
21	1	c.40dup/c.40dup	p.(L14Pfs*35)/p.(L14Pfs*35)	frameshift	NA	NA	[36]
22	1	c.434_435dup/c.704+5G>A	p.(W146Sfs*161)/p.?	frameshift/splicing	NA	NA	[37]
23	1	c.1114-2A>C/c.1114-2A>C	p.?/p.?	splicing	NA	NA	[19,20]
24	2	c.704+1G>A/c.704+1G>A	p.?/p.?	splicing	NA	NA	[19,20]
25	1	c.1258del/c.1258del	p.(E420Rfs*6)/p.(E420Rfs*6)	frameshift	NA	NA	[19,20]
26	1	c.666_670del/c.666_670del	p.(Y222*)/p.(Y222*)	nonsense	NA	NA	[19,20]
27	1	c.862C>T/c.1187+3insT	p.(Q288*)/p.?	nonsense/splicing	NA	NA	[19,26]
28	1	c.1016dup/c.1016dup	p.(H341Tfs*7)/p.(H341Tfs*7)	frameshift	NA	NA	[38]
29	1	c.1385_1390delinsATGCATGCACC/c.1385_1390delinsATGCATGCACC	p.(G462Dfs*34)/p.(G462Dfs*34)	frameshift	NA	NA	[39]
30	1	c.1385_1390delinsATGCATGCACC/c.1385_1390delinsATGCATGCACC	p.(G462Dfs*34)/p.(G462Dfs*34)	frameshift	NA	NA	[40]
31	1	c.361C>T/c.361C>T	p.(R121*)/p.(R121*)	nonsense	NA	NA	[40]
32	2	c.1270C>T/c.1270C>T	p.(R424*)/p.(R424*)	nonsense	NA	NA	[41]
33	2	c.995-1G>A/c.995-1G>A	p.?/p.?	splicing	NA	NA	[42]
34	1	c.1187+5G>T/c.1187+5G>T	p.?/p.?	splicing	NA	NA	[43]
35	1	c.1087C>A/c.1087C>A	p.(H363N)/p.(H363N)	missense/missense	Reduced (F), slightly reduced (LC), increased (P)	Increased (GlcSph); slightly increased (chito)	Present study

^§^ Authors cannot exclude hemizygosity of the variant (due to a wide deletion within chromosome 4), as parental DNA was not available for testing.

**Table 5 ijms-25-06615-t005:** PSAP deficiency in literature. Reference transcript: NM_002778.3; reference protein: NP_002769.1. Description of DNA and protein sequence variants are indicated according to HGVS nomenclature guidelines. Abbreviations: NA = non-assessed, F = fibroblasts.

Family	N° of Patients	Pathogenic Variant	Predicted Protein	Type of Variant	GCase Activity	Plasma Biomarkers	References
1	2	c.1A>T/c.1A>T	p.?/p.?	start-loss	Reduced (F)	NA	[44,45,46]
2	1	c.794del/c.794del	p.(C265Lfs*10)/p.(C265Lfs*10)	frameshift	NA	NA	[47]
3	1	c.794del/c.? ^§^	p.(C265Lfs*10)/p.?	frameshift	NA	NA	[47,48]
4	1	c.1A>T/c.1A>T	p.?/p.?	start-loss	Reduced (F)	NA	[49]
5	1	c.148C>T/c.148C>T	p. (Q50*)/p.(Q50*)	nonsense	NA	NA	[50]
6	1	c.1006-2A>G/c.1006-2A>G	p.?/p.?	splicing	Reduced (F)	NA	[51]
7	1	c.889G>T/c.889G>T	p.(E297*)/p.(E297*)	nonsense	Reduced (F)	Increased (GlcSph and Lyso-Gb3)	[52]
8	1	c.828_829del/c.828_829del	p.(E276Dfs*27)/p.(E276Dfs*27)	frameshift	Reduced (F)	Increased (GlcSph and Lyso-Gb3)	[51,52]
9	1	c.889G>T/c.889G>T	p.(E297*)/p.(E297*)	nonsense	Reduced (F)	Increased (GlcSph, Lyso-Gb3, and PPCS)	Present study

^§^ Hulkovà and colleagues [47] revised the diagnosis of this patient (previously reported as Niemann–Pick type C [48]). Unfortunately, no material for genotyping was available; however, they identified the c.794del variant in maternal fibroblasts.

## Data Availability

The original contributions presented in the study are included in the article/Appendix A, further inquiries can be directed to the corresponding author/s.

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
