# Peer review of "Deficiency of Glucocerebrosidase Activity beyond Gaucher Disease: PSAP and LIMP-2 Dysfunctions"

_ijms, 2024, doi:10.3390/ijms25126615_

Round 1

Reviewer 1 Report

Comments and Suggestions for Authors

This manuscript describes three patients in detail, two with LIMP2 deficiency and a single case of PSAP deficiency. It looks to compare the biochemical profiles in these conditions with neuronopathic Gaucher disease and highlights potential diagnostic markers. It goes on to demonstrate differences in intracellular processing and proposes possible mechanisms of cellular dysfunction in these conditions. The data is interesting and contribution to the understanding of two ultra-rare conditions is valuable. I recommend the manuscript is accepted for publication with the following points addressed:

1.        Table 1:  Biomarkers. Please state the number of controls used to establish reference ranges. There is reference to PPCS levels in the text (raised PPCS in PSAP patient line 120, normal PPCS in LIMP2 buried in text line 191). Please add PPCS data to Table 1 to make it easier for the reader. If there is any additional biomarker data, then consider adding for completeness.  

2.       Table 2: Glucocerebrosidase activity. Please add the number of controls used to set reference range and include activity range in addition to mean ± SD. It is unclear to me how the % normal control activity in calculated in leucocytes as the data doesn’t correspond to activity/mean activity x 100. Please clarify.

3.       Figure 1A demonstrates that GCase in LIMPS2 patients in fibroblasts when treated with EndoH shows complete cleavage suggesting that the GCase protein is immature and retained in the ER. The authors propose a model where no GCase reaches the lysosome.

The authors go on suggest that in leucocytes there is some LIMP2 independent transport of GCase to the lysosome resulting in residual enzyme activity. Do they therefore propose that some of the GCase in leucocytes is processed to maturity and reaches the lysosome. If this hypothesis is correct, then would western blotting of leucocytes from LIMP2 patients be expected to give a pattern showing both immature and mature protein? A repeat of experiment 1A on leucocytes would be a valuable addition. While it may be difficult to obtain further blood from the patients to repeat this experiment, some discussion on further studies to support this hypothesis would be beneficial.

 4.       Table 3 states that all LIMP2 patients showed reduced plasma GCase but the data shows and manuscript states that it is increased in all patients measured (line 230). This is likely an error in the table, please clarify.

5.     The authors highlight that raised PPCS and positive filipin staining in the PSAP patient are concordant with impaired cholesterol trafficking. Given that the LIMP2 patients also showed a positive filipin stain result then would we not expect PPCS to also be raised. Can the authors suggest why PPCS is normal in LIMP2 but filipin is positive in the discussion?

6.     Supplementary data Table S1 gives GBA1 genotypes. Historical nomenclature continues to be used within the field of Gaucher disease, it would be helpful to include this. See below:

Patient 1 : [c.1448T>C, p.(Leu483Pro)] / [c.1448T>C, p.(Leu483Pro)]  (L444P / L444P)  

These recommendations will not require major alterations to the manuscript and should be easily addressed by the authors.

Reviewer 2 Report

Comments and Suggestions for Authors

The manuscript ID: ijms-2989914 by Eleonora Pavan et al. focuses on two major proteins, LIMP2 and PSAP, which are involved in the transport of glucocerebrosidase (GCase) to the lysosome and its activity.  These two proteins are multi-functional, with both cytosolic and organelle functions. The authors work seek to demonstrate that biallelic mutations in both proteins cause deficiency of GCase.  They try to compare the deficiency of LMP2 and PSAP with the actual deficiency of GCase protein caused by GBA1 mutations. The authors provide a sufficient molecular and biochemical evaluation of two patinates who carry biallelic SCARB2 mutations, coded for LIMP2 protein, and one patient with PSAP mutations. It also included eight Gaucher patient and control samples.

The manuscript is commendably well-written, the data and resources have been clearly and thoughtfully organized. However, a few areas could benefit from further clarification and explanation.

  points:

A-      Glucocerebrosidase deficiency” : In the title, figure legends, and the discussion, the authors emphasize the deficiency of the LIMP2 and PSAP proteins causes the deficiency of GCase protein. the GCase, both RNA expression and protein synthesis are normal. The GCase activity is intrepid by the lack of the other two proteins. Please change “Glucocerebrosidase deficiency “ to “deficiency of or reduce Glucocerebrosidase activity” in the title and in the other part of the manuscript.

B-      Please include the Sanger sequencing data for the SCARB2, and PSAP mutated cases.

C-      On page 2, paragraph 4, the authors mentioned mutations in the SCARB2 gene present AMRF symptoms, a condition that shares some clinical features with GD. Please correct the sentence., AMRF is not a clinical feature of GD. Only myoclonic epilepsy was observed in some rare GD type 3 cases.

D-     On page 4, lines 163-165, the authors mentioned, “the protein was completely retained in the ER.” Since the three cases in this manuscript, there are no GBA1 mutations. Therefore, GCase protein is not a misfolded protein.  It is less likely to have a complete GCase degradation in the ER. Please correct the sentences.

E-      The authors correctly mentioned the possible transport of the GCase to the lysosome by non-LIMP2 transporter. They also observed GCase activity in the plasma of the SCARB2 mutated cases but not in the WBC and fibroblast. This data presents possible secretion of some GCase outside the cells. Please explore this possibility in more detail.

F-       The data presented in Figure 1 were scientifically designed and presented well. in the last paragraph on page 4, please exclude the sentence “although a portion of GCase protein was…..”.Also,  about, LIMP2 deficient cells, the sentence “GCase is completely retained in the ER…..) needs to be excluded.

G-      Please add quantification data for Figure 2.

H-     In Tables 3 and 4, is there any reason or explanation for more than 95% homozygosity of the mutations in the SCARB2 and PSAP genes?  

I-        In Figure 3, please change “GCase deficiency” to deficiency of the GCase activity. Also, add an accumulation of GlcCer to the B, C, and D parts.

J-        Please exclude or change reference number 54. This reference is not related to your statement in lines 243-244.

Reviewer 3 Report

Comments and Suggestions for Authors

The work by Pavan and co-workers, aims at exploring the biochemical and sub-cellular phenotypes that result from hereditary GCase deficiencies/dysfunctions. Overall, the authors have measured two well-known GD biomarkers, as well as a few other glycosphingolipidoses’ biomarkers, and assessed intracellular accumulation of unesterified cholesterol by Filipin staining, in two AMRF and one PSAP deficient patient-derived samples. I have no major doubts in considering that the results they gathered are worth sharing, as reports on the biochemical and cytological effects of ‘indirect’ GCase deficiencies remain scarce.

Still, I would like to share some thoughts with the authors, on the way they built their manuscript, because I believe it may be somehow misleading for the readers. I will try to further elaborate on this: when introducing GCase deficiencies, the authors list all three genes whose deficiencies that can result in it. And they state, on the last sentence of the introduction section that they will “present an in-depth characterization of biochemical and cellular features of patients presenting GCase deficiency due to mutations in GBA1, PSAP and SCARB2 genes”. Still, I feel this sentence does not exactly apply to the study Pavan and co-workers present.

Indeed, while they do refer to 8 GD patients (those who harbor GBA1 disease-causing variants), none of those patients was analised by Filipin staining. Furthermore, even though their plasma levels of chitotriosidase activity and GlcSph were assessed, those results are not presented individually, but either as a group (mean +/- SD; see Table 1). And I don’t mean it doesn’t make sense to group all GD patients together, for they are a sort of “positive control”  on the effects of GCase deficiency here. What I do mean is that one would expect, from the description both in the abstract and introduction section, to find parallel assessments of all three genetic defects – and that’s not true. GD patients are used for biomarker analysis (as a positive control, mostly) and never for Filipin staining (where NPC is used as a positive control, instead).

Thus, I would recommend the authors to critically review their manuscript and reinforce the idea that this paper focuses on the two genetic deficiencies that can lead to secondary GCase defects, just like their title implies.

For example, I do see why the authors have treated GD patients differently in the Results section (grouping their biomarker results together, and not listing their clinical data in the case reports section), but I am unsure whether that is obvious for those readers who are not so familiar with these three diseases. Maybe a disclosure paragraph either at the end of the introduction, or right in the beginning of the Results section, would be enough.

Major remarks

General comments

Please consider rephrasing some paragraphs, in order to make it more obvious that the focus of this manuscript lies on the LIMP2- and PSAP-deficiency cases. I will refer to the last paragraph on the introduction section, but it applies elsewhere.

Minor remarks:

General comments

·       Please pay attention to the nomenclature used to identify the AMRF patients. It is not uniform in the text. Sometimes the authors coin their samples as “LIMP_PT” and “LIMP_PT1” and others as “LIMP_PT1” and “LIMP_PT2”. That does not contribute to a correct interpretation of the results, and makes the paper hard to follow for the readers. It is mandatory for the authors to keep consistency throughout the whole manuscript. Please carefully double-check the whole document, figures and tables included.

·       Please revise the use of the term “mutation”. According to the most recent recommendations/literature, we should avoid using the term mutation, and use “disease-causing variant” or “pathogenic variant” instead.

·       Please doublecheck the recurrent use of capital letters when fisrt referring enzymes or activator proteins. Usually we do not write “Saposin” or “Prosaposin” (etc) with initial capitals when embedded in the text.

·       Why do the authors use the “1” in “GBA1” and not the “2” in “LIMP”?

·       On the results section, the authors use the heading “GD biomarkers”. Yet, from reading the subsequent paragraph, we realize this is an understatement, as other biomarkers were also assessed, namely Lyso-Gb3 and PPCS. Thus, I would consider it more adequate/interesting, to refer to these results and “glycosphingolipidoses biomarkers”, p.e. Another possibility would be to create two small sections, instead of a single one (e.g: 2.2.1. GD biomarkers and 2.2.2. Other biomarkers). Also: any particular reason for not listing the results for Lyso-Gb3 and PPCS in the table (T1)? It would certainly be more informative, and easier/faster to track for the readers.

·       From my experience on the IJMS guidelines, and according to the general canon, when a paper has separate sections for the Results and Discussion, one should avoid drawing conclusions/comments on the listed observations in the results section. Instead, one should leave all sorts of considerations to the discussion section. Thus, I would recommend the authors not to write comments such as “were expected” (line 143), “was quite unexpected” (line 148) or “hypothesize an impairment” (line 190) or “These observations suggest an impairment of cholesterol metabolism (…)” (line 199-200).

Other corrections

·       On page 2, line 54 – please rephrase “SapC is a member of four small lysosomal” to “SapC is a member of the saposin family, which includes four lysosomal (…)” or “SapC belongs to a family of four lysosomal (…)”

·       On page 2, line 56 – please include either “a single” or “a common” before introducing the name precursor.

·       On page 2, line 77 – please include the OMIM number for AMRF.

·       On page 3, section 2.1., line 95 – I would suggest a slight rephrasing of Case#1. As it is, it seems that the patient is currently 34 yo and pregnant when, instead, that was her age/condition when her first symptoms arose (or got registered).

·       On page 3, line 114 – please include a brief disclosure on the use of Anysomicin: what was it used for?

·       On page 4, line 134 – please double check the “n” (is it 6 or 8? According to Table S1 and taking into account all that has been said before, there are 8 GBA1 patients carrying biallelic mutations. Why did the authors refer to 6 here?)

·       On page 4, table 2 – while I get the reason why the authors clustered GBA1 patients’ results together, working with mean +/- SD, they did list those patients individually in the Supplementary Material for their genotype data. I would recommend including a similar SM Table with their individual GCase activity results.

·       On page 5, section 2.5. – Elevated plasma PPCS levels are not listed anyware, only briefly referred to in the last sentence of section 2.2. (page 3, lines 124 to 127), whose heading only refers to GD biomarkers, as I have already stressed in a previous comment. Please consider revising this.

·       On page 6, section 3 – a brief introductory paragraph is missing in the Discussion section.

·       On page 6, line 210 – while the authors refer to results on “GD biomarkers” in the literature, there is no reference at all to GD biomarkers or others of any kind in either of the referred tables – only GCase activity is disclosed.

·       On page Tables 3 and 4 – please move the remarks on family#7’ (Table 3) and  #3 (Table 4) genotypes to a table footnote. Also please double-check the numbers on Table 3: aren’t there supposed to be 2 patients included in the last line of the table? Which refers to the present study?

Comments on the Quality of English Language

Overall, even though there are some typos and a few sentences that would benefit from language editing, the paper is quite readable. 
